# Forest Sharing® as an Innovative Facility for Sustainable Forest Management of Fragmented Forest Properties: First Results of Its Implementation

Francesca Giannetti [1,2], Andrea Laschi [3], Ilaria Zorzi [2], Cristiano Foderi [1,2], Enrico Cenni [2], Cristiano Guadagnino [2], Giacomo Pinzani [2], Francesco Ermini [4], Francesca Bottalico [2], Guido Milazzo [2], Lorenzo Massai [2], Alessandro Errico [2] and Yamuna Giambastiani [2,5,6,*]

1 Department of Agriculture, Food, Environment and Forestry, University of Florence, 50145 Florence, Italy
2 Bluebiloba Startup Innovativa s.r.l., Via C. Salutati 78, 50126 Florence, Italy
3 Department of Agricultural, Food and Forest Sciences—SAAF, University of Palermo, Viale delle Scienze ed, 4, 90128 Palermo, Italy
4 Department of Information Engineering DINFO, University of Florence, Via di Santa Marta, 3, 50139 Florence, Italy
5 National Research Council, Institute of Bioeconomy, 50019 Florence, Italy
6 Environmental Modelling and Monitoring Laboratory for Sustainable Development, LaMMA Consortium, 50019 Florence, Italy
* Correspondence: giambastiani@lamma.toscana.it

**Abstract:** The forestry sector in Italy and throughout Europe is going through a critical period due to ongoing natural and anthropological processes, such as climate change and the abandonment of rural areas. These processes lead to a constant fragmentation of properties in small forest parcels, with direct impacts on management capacity. In this framework, new sustainable forest management methods are being tested and are shown to be good practices to oppose the decline of forest ecosystems. Their innovative aspects concern the introduction of a form of shared and circular economy, where management is built on the process, rather than on the product. Their technical activities are based on precision forestry systems and digitalization. The new approach takes into consideration the fact that the woods are an asset available to the whole community, in terms of benefits and protection. Forest Sharing® is an example of the application of shared forest management systems, due to which the owner user benefits from several services and opportunities, such as the advanced monitoring platform and the access to investment funds. After eighteen months of activity, the first results of the application of the new management systems can already be seen. Many aspects need further development, such as case studies concerning the enhancement due to forest certification and new recreational activities. Shared forest management systems have the potential to increase the level of knowledge and awareness of citizens about environmental and territorial issues.

**Keywords:** sustainable forest management; forest ownership; small forest properties; forest parcels; soils impact

## 1. Introduction

EU forests are considered a core part of the new European Green Deal, since they play a significant and important role in the EU's greenhouse gas balance and quality of life. They provide multiple ecosystem services, such as climate change and biodiversity loss mitigation, while increasing the green economy and creating a resilient society [1–7].

However, one of the most important challenges for EU policy makers when trying to implement strategies and policies with active and concrete actions is forest fragmentation, considered one of the main dangers to the preservation of forests and biodiversity [7–11]. In Europe, 60% of forest is estimated to be privately owned, with a high number of people owning small-sized forest parcels with an estimated average size of 12.7 ha [5,12]. This

reaches even smaller in some eastern and southeastern EU countries, where most of the owners hold parcels of around 1 ha [9,13].

Moreover, as pointed out by the work of Agnoletti et al. [14], in Italy, "*human influence has affected extension, density, structure and species composition of Italian forests in all the geographical areas with no dangers of deforestation, but rather an uncontrolled increase of forests, that need of further afforestation but rather to better manage the existing ones*".

Recently, the issue of forest ownership has received growing attention in research and policy since it can impact the reach of EU goals [9,13,15] and increase the abandonment of forest lands. In fact, land abandonment has been identified in recent years as one of the most important local-scale causes of landscape change, especially in mountain and marginal areas depopulated since the end of nineteenth century due to socioeconomic changes as a result of rural–urban migration [14,16].

As underlined in the paper of Mantero et al. [16], the Mediterranean Basin is one of the most likely areas to experience the detrimental consequences of land abandonment, with increasing degradation processes that can also affect soils. In fact, abandonment outcomes could affect disturbance regimes, introducing novel disturbances within ecosystems or modifying the characteristics of existing ones. Land abandonment could raise wildfire risk (increase in forest fuels) [17] and flooding risk (altering hydraulic flow and water balance) [16,18]. Moreover, as reported by Vayreda et al. [19], the capacity of unmanaged forests to act as a carbon sink is reducing due to climate change, which has caused a lower water availability. Dalmonech et al. [20] underlined that active sustainable forest management can act as a nature-based method for carbon sequestration and to slow anthropogenic climate change, as well as supporting the EU's forest-based climate change mitigation strategy.

In Italy, forests cover an area of approximately 11 million ha [21]. Based on the last national forest inventory, 63.5% is classified as private property, with private forest property prevails in almost every region with the exception of Trentino, Abruzzo, and Sicilia. In some cases, the account of private forest increases to 80% in central Italy (i.e., the Liguria, Emilia Romagna, Toscana, and Marche regions) [21]. Furthermore, as pointed out by the report on "the state of Italian forests" ("Rapporto sullo stato delle foreste in Italia"), the current situation of Italian forest resources is highly complex [22] because of the complex environmental variabilities present in the peninsula (see for more details Chirici et al. [23], D'Amico et al. [24] Giannetti et al. [25]), and the absence of local forest supply chains which are well established only in a few local well-organized territories [22]. Moreover, in the Italian Apennines, the rural areas are almost always depopulated [14], with few and mostly elderly residents, and forest management activities are carried out just for wood production and some limited non-wood forest products (e.g., chestnut production) by small family companies that work within a limited area. In this context, the Italian forest strategies [26] points out that one of the problems that needs to be overcome to develop local supply chains that enhance all forest ecosystem services (e.g., biodiversity, soil conservation, forest conservation, wood and non-wood forest products, touristic activities) [7,22,27] is the fragmentation of forest properties and the absence of cooperative activities. For this reason, the Italian forest strategy [26] supports the birth of a new cooperative approach through forest communities that can support the bioeconomy and the green economy of rural areas [5,22,26].

Furthermore, the Italian forest strategies and EU forest strategies have underlined that, to support sustainable forest management, especially in the Mediterranean area, it is important to provide digital solution tools that can support the development of multi-objective forest management plans [28–30]. In fact, easy-to-use digital tools are recognized as pivotal for the value chains of forest bio-economy sectors [28,31], and are useful tools to support the quantification of different forest ecosystem services.

For this reason, it is important to create new ways to overcome the weaknesses of fragmented ownerships and to provide easy-to use digital tools that can be adopted by the Italian forest sector. The main objective of Forest Sharing® is to offer users the chance to take

part in shared forest management systems, where they can benefit from several services and opportunities, including access to investment funds and advanced monitoring platforms. However, before choosing the most appropriate shared multi-objective sustainable forest management (SFM) system, it is important to understand the characteristics of the slopes of abandoned forest lands and some other details (e.g., forest extension, fragmentation, primary and secondary attitudes that forest owners want to give to their properties) [7]. The knowledge on forest ownership characteristics and forest parcels is necessary when choosing the most appropriate sustainable forest management plans and objectives that can impact soils and forests the least. Basic information on forest ownership characteristics and forest parcels, that must be understood when new sharing and circular sustainable development models are applied, will be described in more detail in the next section.

The principles of sustainable forest management (SFM) [32], including sustainable forest operations (SFO) [33], must be applied in order to develop a proper silviculture, maximizing the benefits from forests in terms of ecosystem services while minimizing the negative impacts [7]. In this context, the Forest Sharing® platform was born in Italy in 2020 with the aim to create a network for forest owners, professionals in forestry, forest enterprises, and industries related to the wood supply chain, together with public administrations and any stakeholder in forest management. Forest Sharing® is a facility developed as an online platform that collects from forest owners around the whole Italian peninsula, with the aim of creating an opportunity to restart management of unmanaged and abandoned small forest properties using a sustainable multifunctional collaborative and sharing approach [1–3].

This research aims to understand the location of forests and forest owners registered to the Forest Sharing® platform, in order to better understand the fragmentation in forest ownership and to compare it with the current Italian and European situation. Moreover, once locations were identified, forest characteristics were assessed. It was very important to determine the size of each forest parcel, distance of the forest from the residence of forest owners, accessibility, soil properties, and forest slope. This helped in defining limitations and advantages of different forests spread throughout Italy, and to choose the best forest management practice. The subsequent step was to identify the expectations of the forest owners and identify the attitudes each owner wanted to give to their forest, divided into primary and secondary preferred attitudes. The attitudes were then compared with the different characteristics studied, to better understand the capability of forest owners to identify the potentiality of their properties. Finally, the outcomes will serve the owners in better decision-making about their forests, and help them to identify the preferred attitude needed to better preserve and manage their woodlands. This will help to establish and organize the best management practices for each forest parcel, taking into account the physical and environmental parameters.

Another considerable goal was to raise awareness in the scientific community about the existence of the platform and share the preliminary results, in order for the project to be replicable, if needed, in other countries. Moreover, currently, new advancements of the platform are in progress.

## 2. Forest Sharing® Multi-Objective Shared Sustainable Forest Management Approach

Forest Sharing® proposes a new way to manage forests, based on a circular and shared approach. Starting from the assumption underlined by the new Italian forest regulations and the Italian forest strategy [26], which identify the need for forestry sectors to create new forest communities that will implement multi-objective sustainable forest management, the Forest Sharing® platform was launched on the internet in 2020. The intent was to create an on-line community [34] of private forest owners, forest associations, forest companies, forest industries, and all citizens. The assumptions behind the creation of the Forest Sharing® platform are that forests provide a large number of forest ecosystem services from which the whole community benefits. In the past, this happened in the countryside villages and in

the mountain areas, where the community together managed extensive portions of forests from which they daily received wood and non-wood forest products and services [14].

Currently, as mentioned in the European forest [1], soil [4], and biodiversity [3] strategies, and the Italian forest strategy [26], it is necessary to start to think again about sustainable development models and give a central role to sustainable forest management (SFM), as also reported in the new EU Green Deal [2],. In fact, there is the need to increase the area of SFM since it is recognized as important, not just for wood and non-wood forest products, but also to preserve biodiversity, soil conservation, and carbon stock [1–4,20,27].

The Forest Sharing® approach process starts with the owners who voluntary decide to subscribe to the platform. This allow them to enter into a community where they will network with forest managers, technical engineering, forest and non-forest companies (e.g., harvesting companies, tourist associations, environmental guides), other owners, associations, and other stakeholders. Due to the network, new opportunities arise with the intent that all the action taken will be applied to take care of forests, applying different multi-objective sustainable forest management systems/plans that will preserve the potential of forest to provide multiple ecosystem-services [27].

The circular economy applied by Forest Sharing® to the forest sector is based on the concept that wood and non-wood forest products are just one component within a variety of other activities/services that can be carried out in the forest. Due to the community created by the Forest Sharing® platform, it is possible to combine fire prevention activities (e.g., road planning) with tourism enhancement (e.g., roads can be used for hiking, trekking, and horse activities), or to integrate the income from harvesting with the costs of protecting rare plant species through monitoring systems. This will be possible because of the creation of multi-objective forest management plans that include forests from different owners that will share the economic and environmental income with the neighboring or bordering forests, as has been done in the past.

When forest owners subscribe to the platform, they obtain access to a dashboard showing information provided by GIS analysis, which will be described in detail in the next sections. Navigating the personal area, forest owners can decide to directly make decisions regarding the management of their properties, or they can delegate to other stakeholders networked in the platform (e.g., farms, cooperatives, associations) based on the contract they prefer and on the basis of economic expectations. During the subscription of the platform, the owners' preferred form of management is indicated, which will be validated by Forest Sharing® forest managers based on the characteristics of the forest.

## 2.1. Data

### 2.1.1. Forest Owners Data

Based on the GIS analysis carried out with the Forest Sharing® platform, here we provide the first results of the analysis of the 9585 cadaster parcels (Figure 1) representing a total forest area of 13,981 ha, owned by 684 users of the platform (i.e., private forest owners) in 644 different identified land properties. The analysis was carried out on parcels registered in the last two years (1 January 2021–22 November 2022) on the Forest Sharing® platform. As stressed in many previous papers (e.g., Chirici et al. [23], Giannetti et al. [25,35] D'Amico et al. [24]), access to detailed information on forest parcels is essential for implementing SFM in a multi-objective way. In the results we provide an overview of the characteristics of the forest parcels and their ownership. These form the baseline for the implementation of multi-objective forest management plans and are essential for decision making.

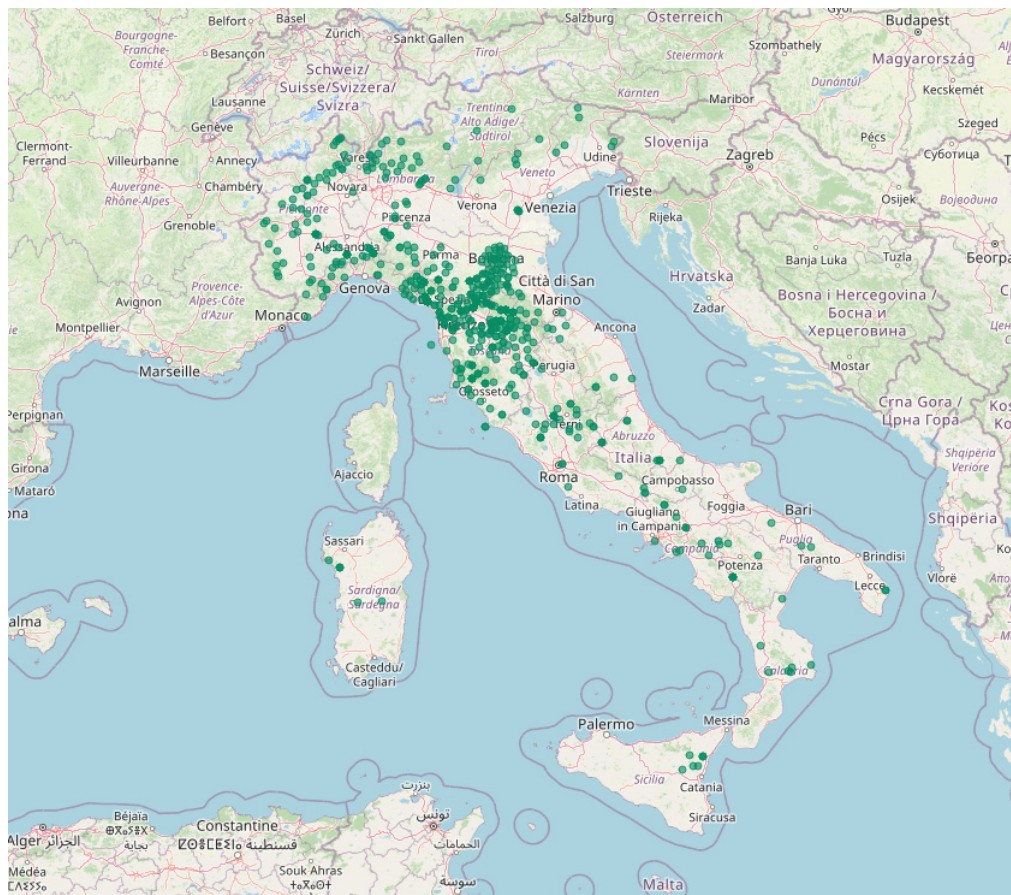

**Figure 1.** Location of the 9585 cadaster parcels subscribed by 684 users in Forest Sharing® platform.

Data on forest owners are directly acquired by the platform during the application process using a dedicated form. The forest owners' data used for the GIS analysis are (1) municipality and province of residence of the owner, (2) cadaster information of forest lands (name of the major, number of cadaster parcels), (3) two main attitudes of the preferable SFM attitudes that can be chosen from the following: A—Productive, B—Conservation, C—Touristic, D—Protective, E—Recreation, and G—Unknown. In Table 1 we reported the details of the description of the main attitude of SFM systems as reported in the Forest Sharing® platform.

### 2.1.2. Open Street Map Streets/Roads Geographic Layer

OpenStreetMap (OSM) has emerged as a global project and community operating with the objective of creating and maintaining a free and editable database and a world map based on the contributions of volunteer mappers [36]. More detail about the potential use of OSM can be found in Grinberg et al. [37]. From the OSM we extracted the streets/roads dataset in the form of vector GIS line features. The line feature vector layers of streets/roads was used to calculate the accessibility of forest parcels subscribed to the Forest Sharing® platform. In fact, the analysis of forest accessibility and forest road network are necessary when choosing the most appropriate harvesting system to lessen the impact on soils [38–40], to prevent forest fires [40], and for tourist and excursion activities [41].

**Table 1.** Description of attitude of sustainable forest management system on the Forest Sharing® platform.

| Attitude of SFM | | Description |
|---|---|---|
| A- | Productive | An SFM system to produce wood and non-wood forest products. Forest operation and design to maximize production and therefore the revenues generated by the forest in accordance with the SFM criteria. The productivity depends on the type of forest, wood species, soil fertility, and climatic factors, but above all on accessibility. |
| B- | Protective | An SFM system that tries to enhances the protective role of forests in relation to destructive natural phenomena and which lead to risks for man, infrastructure, and settlements. The planned interventions will reduce hydrogeological instability, floods, fires, the impact of the wind, etc. The economic resources necessary to implement the interventions derive from numerous sources (private funds, donations, research projects, community contributions). The benefit that can be deducted from the forest corresponds to the ecosystem service performed. |
| C- | Touristic/Excursion | An SFM system that tries to enhance fruitful activities, such as trekking, hiking, trail running, forest therapy and all those outdoor activities that do not require particular structures. The benefit that can be obtained from the forest derives from the use of these areas in relation to the activities carried out. |
| D- | Recreational | An SFM system based on the use/building of particular structures and infrastructures to develop entertainment, sport and leisure activities, such as adventure parks, campsites, glamping. The benefits of the forest derive from the management of these activities. |
| E- | Conservation | Forest management dedicated to the conservation of natural aspects, such as the protection of rare species, maintenance of protected habitats, biodiversity, carbon storage, etc. The benefits can arise from the ecosystem services obtained. |
| F- | Unknown | The owner does not choose any of the attitudes for various reasons, because he does not know the potential and functionality of his forest. |

### 2.1.3. Digital Terrain Models

The digital terrain models (DTMs) used in Forest Sharing® platform are:

(1) LiDAR DTMs available at 1, 2, 3, and 5 m resolutions. The LiDAR DTMs are freely available and were downloaded from regional geographic portals or from the National Cartographic portal. For a more detailed overview of LiDAR in Italy, we refer to D'Amico et al. [24]

(2) The 10 m resolution DEM TINITALY, which is the most fine-scale and most accurate DEM currently available consistently in Italy [42–44]. TINITALY is available at http://tinitaly.pi.ingv.it/ (accessed on 30 November 2022) in grid format.

The DTM are used by the platform to perform analysis regarding the slopes.

### 2.1.4. Land Cover

In Italy, not all regions have forest type maps (see D'Amico et al. [24]). For this reason, Corine Land Cover Level IV was used since it is the only consistent dataset regarding forests covering the whole Italian territory [24,25,45].

The nomenclature system of Corine Land Cover Level IV provided for forest classes is a more precise nomenclature system compared to the standard ones required by the EU. For the class "forest", the nomenclature systems allows the user to distinguish between the following classes: "Forest dominated by holm oak and/or cork oak", "Forest dominated by deciduous oak (Turkey oak, downy oak, farnetto oak, and/or English oak)", "Mixed forests with a prevalence of mesophilic and mesothermophilous broad-leaved trees (maple-ash, cute black-ash)", "Chestnut forests", "Beech forests", "Forests dominated by hygrophilous species (forests with a prevalence of willows, poplars, and/or alders, etc.)", "Forests dominated by Mediterranean pines (stone pine, maritime pine) and cypress", "Forests dominated by mountain and Mediterranean pines (black pine and larch, Scots pine, Bosnian pine)", "Forests dominated by silver fir and/or spruce", "Forests dominated by larch and/or stone pine", "Mixed forests with a prevalence of broad-leaved trees", and "Mixed forests with a prevalence of conifers".

### 2.1.5. Soils Data

The soil data were extracted from the datasets of the Joint Research Center—European Soil Data Center (JRC-ESDAC) [46], which makes available multiple raster datasets with 1 km × 1 km resolution, updated to 2006. The dataset of EU soils contains the following factors at the polygon level: (i) LS = slope length coefficient (dimensionless), (ii) K = soil erodibility factor (dimensionless), (iii) ER = Soil erosion (mg ha$^{-1}$ year$^{-1}$) [46].

### 2.1.6. Aboveground Biomass

In the platform, the "Global Ecosystem Dynamics Investigation (GEDI) L4B" product provides 1 km × 1 km (hereafter 1 km) estimates of mean aboveground biomass density (AGBD) (Figure 2) based on observations from mission week 19 starting on 18 April 2019 to mission week 138 ending on 4 August 2021. This time-scale is used since at the moment it provides the most accurate AGBD data in terms of "nominal years" available at the Italian national level [47] The maps of Italy, the Joint Research Center and the ESA were updated in 2005, 2010 and 2018, respectively [25]. The maps provide AGBD in Mg ha$^{-1}$.

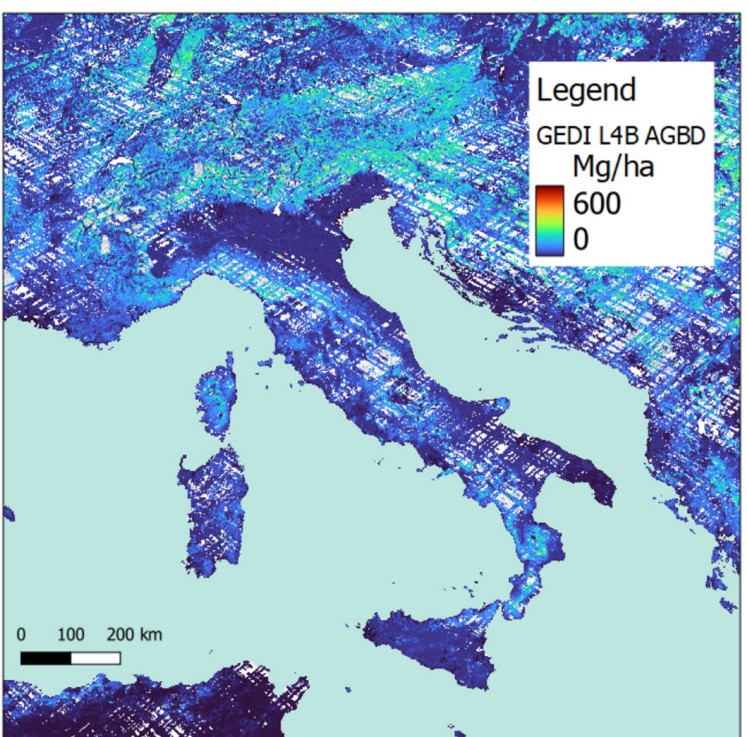

**Figure 2.** GEDI L4B product reporting the mean aboveground biomass density [47].

### 2.2. GIS Analysis

The Forest Sharing® platform is equipped with many automatic GIS analysis tools and processes data based on the available Python "rgdal" and "rgeos" [48,49] libraries.

When a new owner subscribes to the platform, the GIS analysis works to provide two different levels of information: (i) ownership characteristics and (ii) forest parcel characteristics

### 2.2.1. Ownership Characteristics

Firstly, the platform analyzes the number and the size of the forest parcels added by each forest owner in order to calculate the total area per forest owner.

Secondly, using the "GeoPy" function to calculate the distance between two points, we calculated the distance between the forest parcels (centroid of the forest parcel polygon) and the residence of the forest owners (point of location of house). The distance was calculated as a linear distance.

### 2.2.2. Forest Parcel Characteristics

On the basis of the most accurate DTM available, we derived the "slope" in terms of percentage and "exposure" using the "terrain-analysis" tool. For each parcel we extracted the mean, maximum slope, altitude, and the prevalence exposure using the "zonal statistics tools".

As soil is one of the most important components to be conserved and maintained in forest ecosystems, the Forest Sharing® platform collects the data related to soils based on the European Soil Data Center [46]. In particular, the following mean parameters affecting the soil loss are extracted for each cadaster using zonal statistics: LS = slope length coefficient (dimensionless), K = soil erodibility factor (dimensionless), ER = Soil erosion (mg ha$^{-1}$ year$^{-1}$). For each parcel, Forest Sharing® is able to calculate the mean value for these parameters, which will guide the planning of forest operations on the basis of erosion risk. When necessary, the platform can implement models such as RUSLE [50] to estimate soil erosion pre- and post- forest operations, incorporating DTM, land cover (either from Corine land cover or by classifying NDVI derived from satellite images), and rainfall data in order to estimate the impact on soil.

On the basis of the OSM street/road layers, we analyzed the distances between the nearest road and the centroid of each forest parcel. This provides information regarding the accessibility of each parcel, which is necessary for understanding harvesting, forest fire prevention, and excursion and tourist activities.

Moreover, to understand the available AGBD for each parcel, we extracted the mean GDBD value on the basis of the GEDI L4B and multiplied it by the area of each parcel.

## 3. Results

The initial assessment of the forest fragmentation revealed that many forest owners held small forests. In fact, the results of the GIS analysis showed that the mean forest area per user is approximately 20 ha. However, if we remove the 11 users with larger and medium-sized forest properties (>100 ha), the mean forest area per user is 8.2 ha, with a mean area of cadaster parcels of 1.4 ha (standard deviation = 5.3, median values = 0.44 ha). Afterwards, we decided to determine the distance proprietors are living from their forests. Results showed that 35% of the forest owners live further than 50 km from their forest, with 15% living further than 200 km away, 10% living within 20 km, and 40% living within 10 km.

Thereafter, we analyzed the attitude preferences of the forest owner registered to the platform, dividing the primary attitude selection from the secondary. From a total of 644 identified land properties on the platform, 316 owners chose a first SFM attitude of "Productive", 160 chose "Unknown", 63 chose "Protective", 47 chose "Touristic/Excursion", 38 chose "Conservation", and 20 chose "Recreational"(Figure 3).

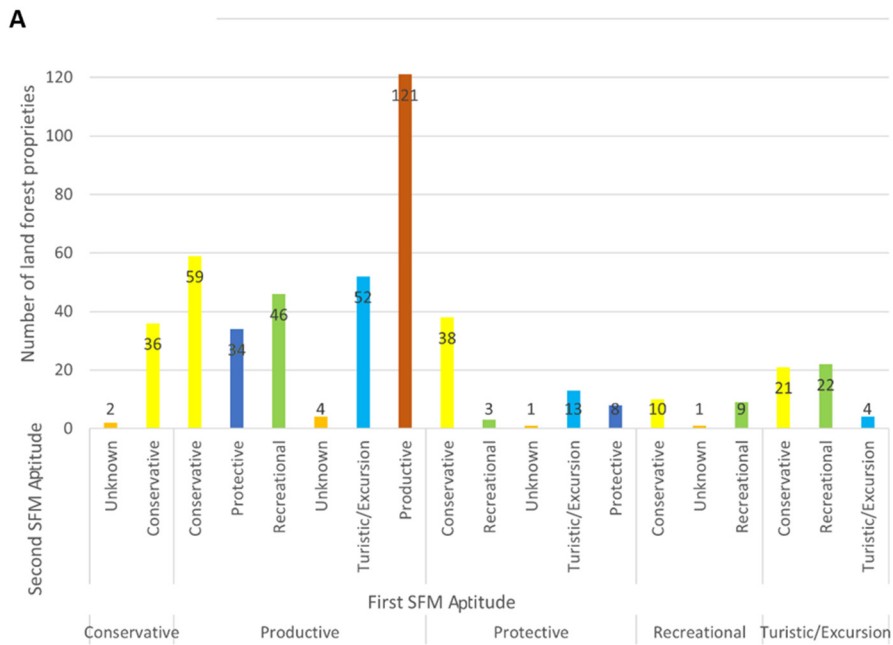

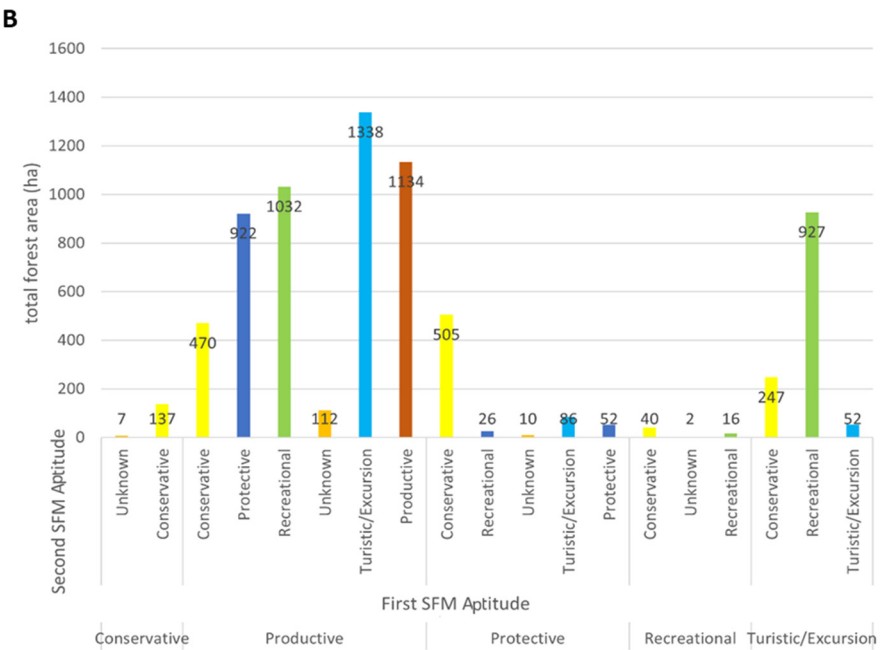

**Figure 3.** Panel (**A**). Number of forest properties per primary and secondary SFM attitude. Panel (**B**). Total forest area (ha) per primary and secondary SFM attitude.

Of the total 316 users who selected "Productive" as the primary SFM attitude, 121 users did not select any other secondary SFM attitude, while 59 selected "Conservation", 52 selected "Touristic/Excursion", 46 selected "Recreational", 34 selected "Protective", and 4 selected "Unknown". In Figure 3 it is possible to see that the owners who did not choose "Productive" as the primary SFM attitude did not choose "Productive" as the secondary SFM attitude either.

Moreover, the "Productive" SFM attitude represents a total of 5008 ha, the "Unknown" SFM attitude represents 4914 ha, "Touristic/Excursion" 1226 ha, "Protective" 676 ha, "Conservation" 144 ha, "and "Recreational" 58 ha (Figure 3).

From Figure 4 it is possible to see the distribution of land forest properties among the different SFMs. Results on forest extension show that the larger forest land properties (total forest area > 100 ha) represent a total of 5925 ha, owned by 11 users (Figure 4-Panel A); the medium-sized forest land properties (total forest area between 50 ha and 100 ha) represent a total forest area of 1968 ha, owned by 16 users (Figure 4-Panel B); the small forest land properties (total forest area < 50 ha) represent a total of 4136 ha, owned by 409 different users (Figure 4-Panel C). From Figure 4, panel C, it is possible to see that the mean values of forest area for small land forest properties are always smaller than 5 ha.

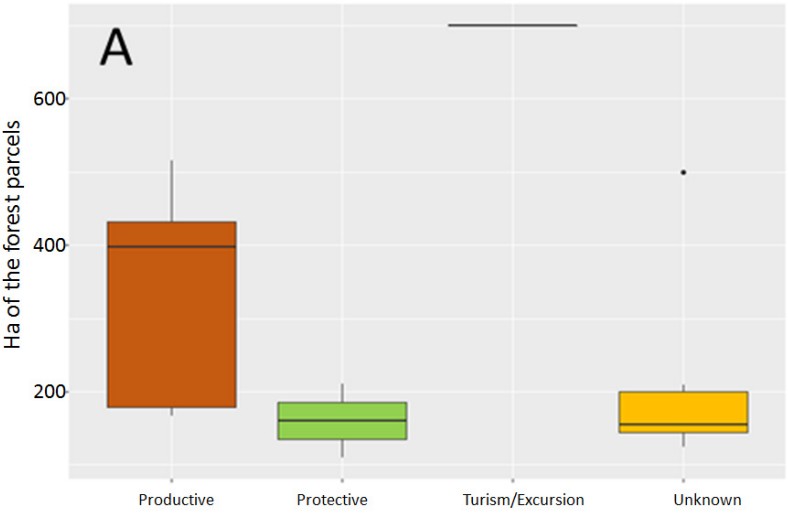

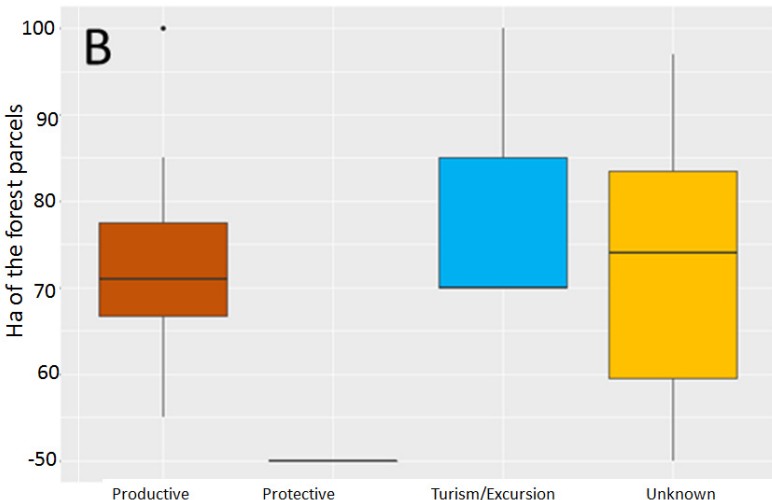

**Figure 4.** *Cont.*

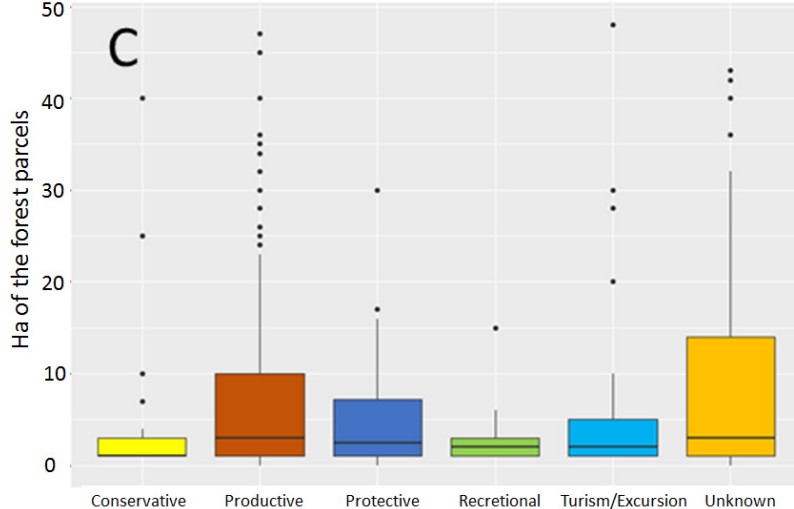

**Figure 4.** Boxplot representing the distribution of the area of forest properties per first SFM attitude Panel (**A**). For larger forest proprieties (total forest area $\geq$ 100 ha). Panel (**B**). for medium forest properties (50 $\leq$ total forest area < 100 ha) Panel (**C**). For small forest properties (total forest area < 50 ha).

　　　Analyzing the distribution (Figure 5) of K, ER, and LS, representing soil erosion, we can see that: the values of K ranged between 0.018 and 0.049, with a median value of 0.029 and a mean value of 0.030; values of LS ranged between 0.036 and 52.87, with a median value of 4.23 and a mean value of 4.63; ER ranged between 0.0047 and 103.89, with a median value of 0.47 and a mean value of 6.08; and slope ranged between 0 and 55%, with a mean value of 15%. From Figure 5 it is possible to note that the highest mean of K occurs for the parcels with the chosen SFM attitude "Unknown", followed by "Productive", while the lower values were observed for the "Recreational" attitude. A high mean of LS was observed for the SFM attitude "Conservation", while high value of ER were observed for "Recreational", and of slope for "Conservation". For the AGBD, the values ranged between 4.04 Mg/ha and 239.41 Mg/ha with a median value of 100.54 Mg/ha and a mean value of 101 Mg/ha.

　　　Analyzing the correlation between the LS, ER, K, and ABGD extracted on the basis of cadaster parcels we found that the Pearson coefficient of correlation *r* was between −0.33 and 0.35, with significance always high (*p*-value < 0.001) (Figure 6).

　　　Analyzing the accessibility of the forest cadaster parcels (Figure 7), we can see that there are many parcels without good accessibility (distance from the road higher than 500 m), and that the mean values for all the SFM attitudes remain within 250 m.

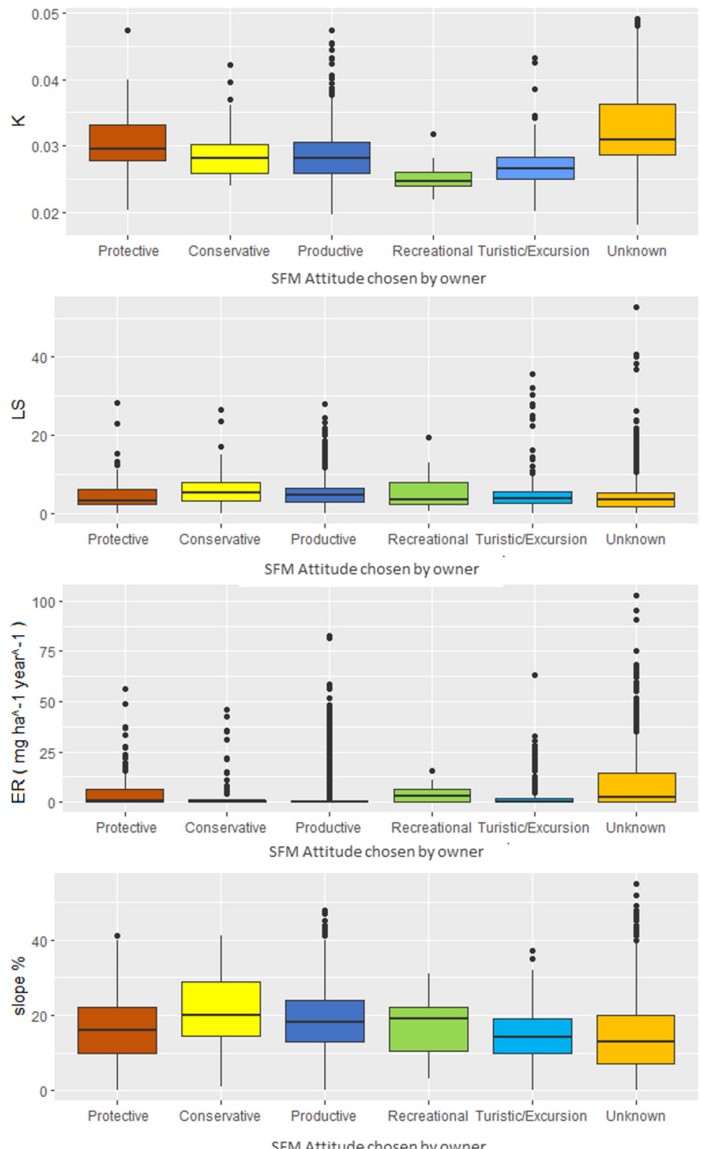

**Figure 5.** Distribution of values for K, LS, ER, and slope according to primary SFM attitude of parcel in Forest Sharing® platform.

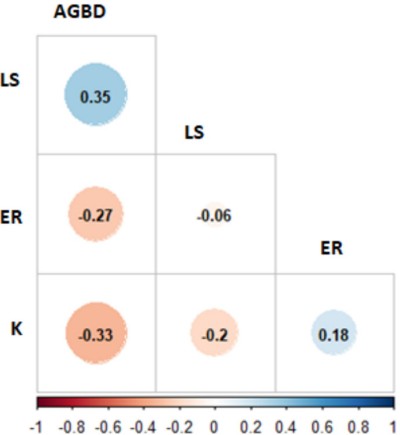

**Figure 6.** Correlation analysis between the LS, ER, K, and ABGD values at parcel levels. All the reported correlation coefficients are calculated with the 95% significant level and have *p*-values < 0.001.

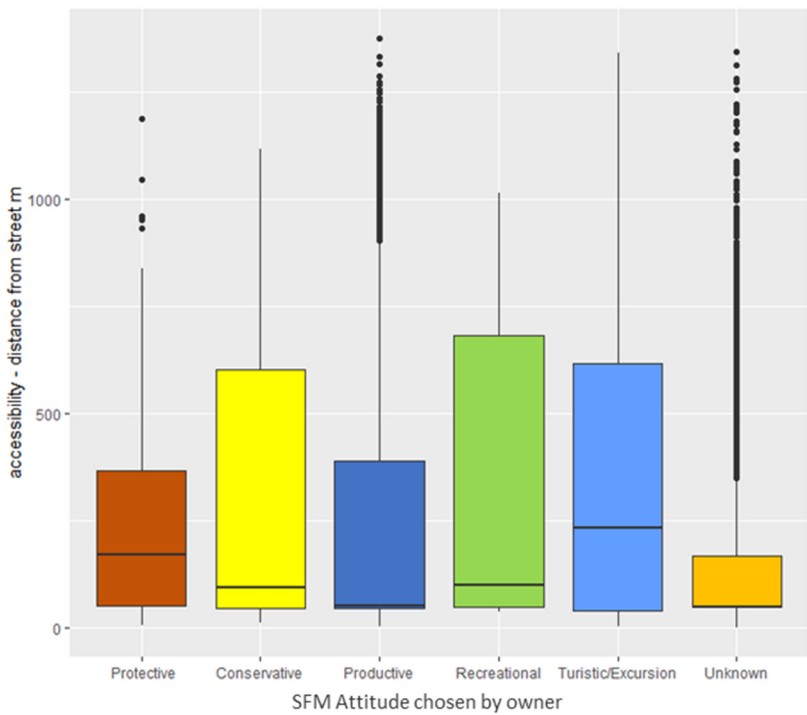

**Figure 7.** Distribution of accessibility by first SFM attitude for parcels in Forest Sharing® platform.

## 4. Discussion

The Italian forest scenario, based on the data collected via the Forest Sharing® platform, is confirmed to be very fragmented, as demonstrated by other studies on European forests [6,8,11]. Moreover, the properties are small, with a mean forest area per user of 8.2 ha, which is less than European average (12.7 ha) [13,16], and are often abandoned. This dynamic can leads to a degradation of forest resources [14,19,51,52].

In just 2 years, Forest Sharing® has brought together many forest owners (i.e., 644 forest owners) who are unable to manage their forest by themselves and who need technical support. Lack of knowledge of forest management in small privately owned forests has been assessed by other studies examining [53]. Another problem clearly evident from the data collected with the platform concerns the scarce forest road network, which makes forest operations very difficult and very expensive, both for productive, conservation and touristic purposes [38,39,54]. It is also important to point out that it is not possible to understand the state of forest roads and their dimensions from the OSM road/street network. Therefore, based on our knowledge, we can suppose that the accessibility could be worse than we estimated, especially in the Apennine area and in the south of Italy, where forest roads have not been well maintained over the last 50 years [14] and where there is also a large presence of private forests according to national forest inventory data [21].

The GIS analysis illustrated in this paper allowed us to understand some important socio-economic data regarding small forest properties. First of all, it is important to underline, as mentioned by other studies [14,52], that many forest owners live far from their properties (>50 km), resulting in them being less interested in managing their forests, as pointed out also by Quiroga et al., (2019) [55]. Moreover, when Forest Sharing® staff spoke with some of the owners through direct interactions, they reported that in the 75% of the cases they were not aware of their forest location since they inherited it from relatives. For this reason, the platform is of key importance for giving small forest owners some important information regarding the characteristic of their properties using available consistent data and allowing them to make some technical considerations [24]. Moreover, the information shared in the platform gives the small forest proprietors the possibility to become active players in SFM, since forest owners can start to know their properties better, and can reconnect with forest culture [55].



Most of the Forest Sharing[®] forest owners indicated "Productive" as their first SFM attitude, especially for larger and medium forest properties. This underlines forest revenue as a high priority, as confirmed by Feliciano et al., 2017 in other EU countries [56]. Thus, it has become necessary to understand the impact of forest operations on soils to mitigate the possibility of soil loss. To give forest owners the perception on how SFM and SFO can have an impact on soils, Forest Sharing[®] provides this information on all the users' dashboards. As mentioned by previous studies, the analysis of RUSLE allows the monitoring of the state of soils [57–59]. Hence, Forest Sharing[®] parcels are classified on the basis of soil erosion (ER), and the forest managers of the platform can use this rating to evaluate the applicability of attitude preferences expressed by the owners, and whether some soil and water bioengineering works are needed [60,61]. In fact, in cases of high erosion risk, the Forest Sharing[®] platform proposes to forest owners some mitigation actions, looking also to external funding to preserve soils. Additionally, the K and LS factors are considered when forest operations are carried out. For example, if the value of the K parameter is high (greater than 0.04, [62]), we propose to reduce the forest covers, since erosion in this case depends on the soil characteristics alone, while, when LS is high, Forest Sharing[®] proposes forest operations as small non-continuous cuts along the maximum slope, since this erosion is mainly due to the morphological characteristics of the slope. Due to K and LS, the cut can be planned in an proper way [57,59]. Moreover, since we have the possibility to calculate the accessibility of the parcels, we can also plan new roads where needed and also use the data to calculate the most accurate forest harvesting system, as carried out in previous work by Laschi et al. [38].

The methodology proposed with the Forest Sharing[®] platform makes it possible to develop a new approach for SFM, where not only the environmental sustainability of the interventions is taken into account, but also the generation of economic, social, and environmental benefits in general, obtained due to a balanced use of forest resources. The platform makes it possible to develop solid supply chains, introducing innovative monitoring processes [58] and tracking of the supply chain [59]. Wood is a fundamental material for the economic development of Italy and Europe [1–6], and must be produced on a local scale by involving the population in the forest process to improve resilience, particularly in view of ongoing climate change [15,22,47,52]. The owner can express his preference on the attitude and thus maintain his ethical goals and principles. When the aggregation is formed, technicians and companies take advantage of the scalability of the operation, due to large surfaces being subject to planning. The management objectives are in balance with the characteristics and needs of the territory. The results obtained during the first period of activity of the platform allow us to show the feasibility of the Forest Sharing[®] platform approach and the development potential due to the increase in the participation of new users.

## 5. Conclusions

Because of the importance of forest ecosystem services, it is necessary to increase the implementation of sustainable development goals on forests worldwide. In this context, Forest Sharing[®] works to solve a long-standing problem: forest fragmentation, which is a typical effect of the abandonment of remote areas, causing negative effects in terms of forest management, and the digitalization of the sector. While in some European countries (e.g., Germany, Romania, Latvia, Czech Republic) policies are in place to support the creation of forestry associations to overcome fragmentation and small forest SFM [9,63–65], in Italy, this is still lacking. Because of this, it is required by the Italian forest strategy [26], and thus by Forest Sharing[®], that bottom-up initiatives lay the foundation for an innovative approach of association, with encouraging results. In fact, the population of forest owners— according to the showed first results—seems to be reacting to our new strategy. However, it is important to note that out of a total of 6 million ha of private forest in Italy, the Forest Sharing[®] platform has intercepted in two years just 0.2% of forest private areas, so a lot of work needs to be carried out over the next years to have a real impact on the sectors.

However, Forest Sharing® has the potential to become a meeting place for the forest sector, through which owners find companies and technicians to manage their forest or join projects for enhancement and conservation. Due to the web GIS-based structure and other related tools, it can be equipped with software, analysis models, and decision support systems to make forest management more efficient, sustainable, and safe. The creation of a large community of forest owners can play a key role in enhancing ecosystem services and the related monitoring processes, also directly involving citizens.

**Author Contributions:** Conceptualization, F.G., Y.G., L.M. and G.M.; methodology, F.G. and Y.G.; software, F.E., G.P., C.G. and L.M.; validation, F.G., Y.G., C.F., F.B., A.L. and A.E.; formal analysis, F.G. and Y.G.; data curation, F.G., Y.G., F.E., G.P., C.G. and L.M.; writing—original draft preparation, F.G., Y.G. and A.L.; writing—review and editing, I.Z., C.F., A.L., A.E., G.M., F.B. and E.C.; project administration, G.M. and L.M. All authors have read and agreed to the published version of the manuscript.

**Funding:** This research received no external funding. All funding was internal of Bluebiloba Start-up Innovativa SRL.

**Data Availability Statement:** Data is unavailable due to privacy restrictions.

**Conflicts of Interest:** The authors declare no conflict of interest.

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
