# Peer review of "Forest Sharing® as an Innovative Facility for Sustainable Forest Management of Fragmented Forest Properties: First Results of Its Implementation"

_land, doi:10.3390/land12030521_

Round 1
Reviewer 1 Report
The manuscript is an interesting and innovative piece of work, and I thank you for the hard work done by the Italian authors. However, the authors have ample scope to improve the manuscript, and my comments are as follows:
Abstract: At the beginning of the abstract too many backgrounds information and the reader may confused about the objective. Therefore, I suggest tying the background information to the objectives of the study. In addition, on Page 1 Line 41, the authors say: "Systems can already be seen." Which systems? needs to be specific.
The objective of the paper is wide, needs to be specific, and needs to be narrowed down for worldwide readers.
Conclusion: the authors may suggest a future direction for reader that means how this study would be useful for other parts of the world based on the first results.
Based on above comments the manuscript will minor revision.
Good luck
Author Response
dear reviewers,
Thank you very much for the thorough revision work you have done.
we attach a document in which we report the answers comment by comment.
Also find the document in word and pdf with and without tracking changes.
Greetings

Reviewer 2 Report
Dear authors,
I have carefully read the article several times. It presents an innovative online platform for collaborative, forest management decision-making for small private forest owners. Regardless of the exciting topic, I believe your manuscript needs significant improvements. My major comments are below.
1. The title is interesting, but it is unclear why the authors refer to Forest Sharing as a "facility' . The facility is mentioned only in the title, never in the main text. In fact, from the manuscript, it is unclear what Forest Sharing is - an application, program, platform, etc. This should be clearly specified in the manuscript.
2. Redline is missing: From the introduction, it seems that the problem authors are addressing is fragmented forest property. Yet, this issue is brought up in the middle of the introduction, while the first part lists the importance of forests. If the main addressed problem is unmanaged fragmented forests, I would suggest refocusing the first part of the introduction around that theme, reflecting on the relationship between forest management, undamaged forests, and sustainable forest management.
Then, the authors should clearly present the objectives and /or research questions they are addressing from the manuscript. Currently l. 84-85, l.103-107 reads like objectives (although they are unrelated for now), but there is also a section (wrongly) numbered 1.3 titled Objectives. This section actually does not present objectives but methods (or even results), as it explains GIS analysis. Section 1.1. on Forest Sharing could then e made a separate section 2, as it gives background on the matter, or should be presented as part of results.
3. Results seem to present rather descriptive statistics and some correlations, without any comparison fo different scenarios made by Forest Sharing. The amount of text is very modest, and mostly a summarization of the figures provided. It is hard to assess results in more depth without clear objectives in mind. Authors should first clearly define objectives and then present results in relation to them.
4. Discussion is very short, and not really related to results or introduction. It mostly referees to GIS analysis, which is currently actually presented in the Objectives section. It is very confusing. Another paragraph in a few sentences relates back to most dominant forest owners' attitudes, but more reads like results than the discussion. Authors should here discuss their results in relation to other relevant literature, especially relating to using of digital technologies in forestry, collaborative, multi-stakeholder forest management planning, and soil quality. Shortcomings of the study should also be adressed.
5. Conclusions are extremely short, composed of one sentence. Authors should here derive conclusions on their objectives/research questions, discuss future application and relevance for forest sector and sustainable forest management
6. Use of terminology should be consistent (e.g EU, Eu, Europe), and language should be checked and edited. Authors should avoid one or a few -sentence paragraphs.
Author Response

(The authors gave the same response as above.)

Round 2
Reviewer 2 Report
Dear authors,
thank you for revising and improving your manuscript and for compliments on the innovative platform. Although the manuscript is improved, I believe it needs further revision in order to improve the structure.
Although the manuscript is improved, it is still not clear enough what is the main objective of the manuscript (and not the platform itself), and what research gap it is addressing. It seems that the problem statement focuses on sustainable forest management, problems of private forest management (such as fragmentation), and to some extent soil quality. The connection between soil quality, fragmented private forest ownership, and sustainable forest management is not so evident from the introduction. Even more, the review of already existing tools (i.e. applications, and information systems) for sustainable forest management and how they take into consideration aspects of soil quality, or ownership structure seems to be missing.
While the platform seems innovative, it is obviously related (if not limited) to the Italian context. I believe more details should be presented as the background to enable readers to understand the Italian forest sector's structure and problems, and the presented platform's innovativeness and usefulness.
Thus, the manuscript could greatly benefit from further improvements (as outlined in the detailed comments), especially in the first part. I provided my detailed documents in the attached field.
Kind regards

Author Response
dear Reviewer,
Thanks for your review.
here in the attached PDF you will find the answers to your comments.
